# Get the happiness out–An online experiment on the causal effects of positive emotions on giving

Anja Köbrich Leon[1]* , Janosch Schobin[2]

**1** Institute of Economics, University of Kassel, Kassel, Germany, **2** Institute of Sociology, University of Kassel, Kassel, Germany

☯ These authors contributed equally to this work.
\* anja.koebrich@uni-kassel.de

**Data Availability Statement:** All relevant data are within the paper and its Supporting Information files.

**Funding:** Anja Körich Leon: Financial support for this project by the German Federal Ministry of

## Abstract

Our study provides new evidence on the impact of positive emotions on voluntary contributions to public goods in a natural setting. Using a lab-in-field experiment, we induce positive incidental emotions randomly and analyze their effects on donations using a dictator game with multiple beneficiaries. Although we find no significant overall effect of positive incidental emotions on donation levels, our results show a significant shift in the allocation of donations towards the host charity. These findings shed light on the complex role of emotions in donation behavior and can provide valuable insights for organizations seeking to increase charitable giving.

## Introduction

Charities often grapple with the question of how to attract new donors, and one potential solution is to tap into emotional engagement. This approach centers around investigating the causal connection between emotional engagement and charitable giving, which is studied from various angles. Of particular interest are the effects of anticipated emotions (1) and emotions felt immediately before or during the act of giving, which have been extensively researched. The prevailing hypothesis in this area is that giving or avoiding to give generates positive or negative emotions, which can affect both (a) the giving decisions of potential donors, as these emotions may be anticipated, and (b) the emotional state of potential donors prior to making donations. In contrast, less is known about how *incidental* emotions [1] affect charitable giving. Incidental emotions are defined as emotions resulting from stimuli that are not directly related to the decision to give. The primary hypothesis in this area suggests that incidental emotions are subconsciously interpreted as cues that either promote or inhibit donations.

Our study examines the later in a "lab-in-field" setting [2]. It is closely related to the lab experimental settings used by Drouvelis and Grosskopf (2016) [3] and Ibanez et al. (2017) [4]. These authors study the impact of incidental emotions on contribution behavior. While the former shows that angry individuals contribute less, on average, than those who are feeling

Education and Research (BMBF) under the funding priority "FONA 3 - Research for Sustainable Development". under grant agreement No. 01LN1708A is gratefully acknowledged. Both authors received the support. For further information please refer to: https://www.bmbf.de/bmbf/de Jansoch Schobin: Financial support for this project by the German Federal Ministry of Education and Research (BMBF) under the funding priority "FONA 3 - Research for Sustainable Development". under grant agreement No. 01LN1708A is gratefully acknowledged. Both authors received the support. For further information please refer to: https://www.bmbf.de/bmbf/de.

**Competing interests:** The authors have declared that no competing interests exist.

happy, the latter demonstrates that incidental emotions do not prompt donations per se, but that the emotion of "awe" can increase the amount given among those who donate.

Our analysis focuses specifically on positive incidental emotions, such as joy and happiness, which we interpret as basic positive emotions [see 5]. Due to the "lab-in-field" setting of our study, it was not possible to elicit negative emotions or compare the effects of negative and positive emotions. For obvious reasons, the host charity—i.e. the charity that hosted our experiment—was not interested in the risk to be associated with a bad experience. To induce positive incidental emotions, we employ priming techniques. Priming relies on a person's unconscious reaction to a specific stimulus. Specifically, we use autobiographical recall and visual stimuli to induce positive incidental emotions [6]. As a novelty, we examine affective primes that promote the sharing of positive emotions opposed to just inducing them. This follows the rationale that emotions often serve a social regulatory function, when expressed [7, 8]. Sharing positives emotions within social interaction can be understood as a relationship-maintaining and trust-building mechanism [7, 9]. It should therefore enhance the inclination to trust a charity in a donation context. To study the effect of different affective primes, the participants are randomly assigned to five treatment groups. They are asked to either recall a positive emotionally charged autobiographical event in their past, express it in writing, or take some time to look at an emotionally charged picture or a combination of these stimuli.

The present study is situated within an online lab-in-field setting, driven by the increasing popularity of charitable donations made through online platforms. However, despite the growing trend, empirical evidence from field settings is still scarce [10–14]. Moreover, we analyze the effects of positive incidental emotions on giving behavior in a multi-charity context. When fundraising on online platforms, charities often compete with similar organizations. The presence of appeals for multiple causes can result in receiving fewer donations for each purpose [15]. Consequently, emotional engagement techniques that are not directly linked to donations may pose a potential risk of spillovers to other charities. Against this background, we investigate whether fundraising campaigns that elicit positive incidental emotions attract new donors or whether these positive emotions create spillover effects for adjacent organizations. To measure this, we employ a modified dictator game. It gives the participants the choice to donate an endowment they earned for answering the survey to one of two charities or to keep it for themselves.

Although, we mainly provide null findings on the overall treatment effects on contributions in the dictator game, we also find evidence on the positive effects of two treatments on the probability of giving money to the host charity at the expense of the alternative charity. However, this evidence is not robust for an adjustment for multiple hypotheses using the Bonferroni Correction but retains marginal statistical significance for the less conservative False Discovery Rate (FDR) correction.

Thus, the contribution and main takeaway of this paper is twofold: First, we do not find evidence that incidental positive emotions play a significant role in eliciting donations. Second, employing affective primes that are based on sharing positive emotions might be worth a try to capture a greater share of donations when competing with alternative charities. However, more solid evidence seems warranted to express a strong belief in this assertion. Finally, our explorative results suggest that the interaction of underlying individual motivations that drive individual donations with incidental positive emotions also warrants further study. We find heterogeneous treatment effects depending upon different levels of environmental concerns. This aligns with previous research that suggests the effectiveness of priming techniques is contingent upon an individual's level of psychological engagement with a given charity [16].

The remainder of the paper is organized as follows: Initially, a brief review of relevant literature is presented, and the contribution of the study is outlined. Next, the hypotheses that guide

our research are introduced. The experimental design is explained in detail in what follows. Following that, the subject pool is described and basic results of the study are presented. The findings on heterogeneous treatment effects are then discussed. The paper continues by discussing the practical implications of the results for organizations and finally discusses the findings and concludes.

## Related literature

In the literature, a widespread consensus exists on the relevance of emotions for pro-social behavior [17–20]. More specifically, emotion-inducement experiments tend to show that feelings of happiness make individuals more generous. Primary evidence stems from laboratory experiments [1, 3, 21–23]. Less evident is a negative relationship [24, 25]. Ibanez et al. (2017) fails to find an effect of incidental emotions on giving itself in a laboratory experiment with NGOs as recipients. However, the study shows that the emotional state of "awe" increases the amount donated. Thereby the effects of happiness seem to depend on the economic game used (see 26 [26] for a summary). Field experimental evidence on the role of emotions in charitable giving, however, is rare [27–29]. For instance, Andreoni et al. (2017) use an emphatic stimulus "verbal ask" to elicit empathy, thereby stimulating charitable giving. They find that applying emphatic stimuli both hampers and enhances charitable giving because it increases avoidance as well as generosity. This finding highlights the importance of situational context in field settings. The effect of emotions on giving in field situations is moderated by situational and individual factors, such as avoidance opportunities, as well as by differences in the readiness to be emotionally stimulated, which are usually absent or mitigated in pure lab studies. We provide experimental evidence in an online field setting ("lab-in-field") of the relevance of affective primes as behavioral tools for attracting donors in the context of charitable giving via online platforms. The lab-in-field approach is a recent development that bridges the gap between traditional lab experiments and field experimental data because it "combines elements of both lab and field experiments in using standardized, validated paradigms from the lab in targeting relevant populations in naturalistic settings" [2]. By providing precise, causally interpretable results, this approach can inform practical applications in the field and offer a deeper understanding of donor behavior in naturalistic settings.

When considering the role of emotions in attracting new donors, the economic literature traditionally refers to the effect of anticipated emotions. It is broadly accepted that prosocial behavior produces happiness and individuals engage in it because they anticipate a positive emotional state from doing so [26]. The anticipation of emotional rewards can also affect their emotional state prior to the donation. Wichman and Chan (2022) [30], for instance, find that online donations are anteced by an emotional "preheating" phase in which positive emotions become more salient. Similar arguments are put forward for the effect of negative emotions. Andreoni et al. (2017), for instance, argue that eliciting empathy through a verbal ask triggers guilt if participants do not give. Individuals who anticipate such guilt attempt to avoid the asking situation altogether, whereas participants who engage in the situation give more (often) to avoid guilt and harvest feelings of high status and warm glow [27]. This strong focus on privately experienced emotions, and therefore on the self-regulatory effect of anticipating and processing emotions, however, falls short. It omits the social regulatory functions of emotions, if expressed [7, 8]. According to social-functional approaches in social psychology, emotions play an important role in regulating social interaction [7, 9]. Sharing feelings of happiness with or communicating positive experiences to others constitutes a relation-maintaining and trust-building mechanism [31], regardless of whether the communication partner is represented by a machine [32]. The communication about one's emotions represents an act

of self-disclosure that reduces the social distance between partners and establishes as well as deepens positive social bonds. Through relying on the social regulatory function of emotions when communicated, this paper contributes experimental evidence on the relevance of inducing a social link between the potential donor and the recipient to enhance donations [33–36]. In addition, we expect that elicited emotions that drive social processes motivate respondents to be more inclined towards pleasing the host organization. Thus, showing how behavioral tools aimed at reducing social distance can bias decision-making provides valuable insights for designing experimental studies. For instance, such insight can help to better understand the experimenter-participant interaction in online experiments and how experimenter demand effects can be mitigated in online lab-in-field settings [37].

Research has demonstrated that evoking emotions can lead to increased donations for a single charity. However, it remains unclear whether this effect also holds true in a multi-charity context. Consequently, this paper contributes to the literature on the impact of donation competition between charities on giving behavior. Answering the question of whether fundraising activities increase the total amount of donations or merely redirect donations towards other causes is critical from a charity's perspective [38]. A substitutional relationship of giving, in which the desire to give is in principle satisfied by donations to any charity, may question the general effectiveness of fundraising for increasing donations to a particular public good. The literature so far provides mixed findings on the impact of competition for donations among multiple nonprofit organizations offering similar public goods on giving [15, 38–40]. To this literature, we contribute an experimental study of treatment spillover effects of eliciting incidental emotion as a fundraising strategy. In particular, we analyze whether the treatments increase giving to the host charity (i.e., a green crowdfunding platform) or rather to another charity (i.e., an environmental organization).

## Hypotheses

Existing laboratory evidence on the direction of the causal effect of happiness on generosity tend to reveal a positive link [1, 3, 21–23]. Induced happiness was shown to increase generosity and decrease selfishness. Consequently, we derive the following alternative hypothesis:

> Hypothesis 1: Treatments that induce positive emotions, in comparison with the control condition, increase participants' likelihood of contributing.

As we argued before, the main scientific contribution of this paper is to examine the social regulatory function of positive emotions for enhancing donations to the host organization. We expect different treatment effects depending on whether emotions are socially shared or experienced privately, as self-disclosure, through the communication of emotions, can be seen as an essential social bonding mechanism. Applied to the context of charitable giving in the dictator game, we expect the sharing of positive emotions by the dictator to create a positive bond between the dictator and the host charity. As a result, the dictator's willingness to be generous to the host charity should be strengthened, as it becomes more socially close to the dictator in relative terms. We thus predict:

> Hypothesis 2: Treatments that promote the sharing of positive emotions with the host organization (and induce positive emotions), in comparison with the control condition, increase the participants' likelihood of contributing to the host organization.

## Experimental design

The data for our econometric analysis were collected by means of an online lab-in-field experiment in Germany, which was carried out in cooperation with a crowdfunding platform. This crowdfunding platform pursues a donation-based crowdfunding model and is unique in that it primarily focuses on ecologically sustainable projects. Its user base is highly educated, has a high level of environmental awareness, and strong preference for the well-being of others (see the sample description).

The study was registered at the AEA RCT Registry as trial AEARCTR-0003269 and approved by the German Association for Experimental Economic Research (No. mZkmivhj). We designed a three-part online experiment among visitors to the crowdfunding platform. Fig A.1 in the S1 Appendix offers a visual representation of the experimental design. Data was collected directly from the website of the crowdfunding platform between August 2018 and February 2019. During this period, all visitors to the crowdfunding platform were presented with the opportunity to participate in the survey through a slider on the main page. Before participants were allowed to take part in the study, relevant background information about the study was given and written consent to participate in the study and their agreement to the privacy policy was requested. To avoid recruiting participants more than once, each visitor was assigned a unique visitor identification (i.e., a saved browser cookie) on their devices at the time of their first visit to the platform. Participants who returned to the website later after completing the questionnaire were no longer offered the surveys. As a caveat, we must note that the survey, including the experiment, was offered again in the case of cookie deletion caused by individual browser settings. Among the visitors to the platform, a total of 1041 subjects participated, of whom 1010 completed the survey. Among them, 1008 were successfully assigned to treatments. Completion took an average of 10 minutes. Respondents received €5 in exchange for their participation, which was presented as outside money from the researchers' account. This quid-pro-quo framework was used to mitigate the emergence of positive feelings triggered by an unexpected financial gain.

### 1st Part–Online survey

First, participants take part in an online survey on motivations for donation-based crowdfunding. Based on Bekkers and Wiepking (2011) [41] review of mechanisms that explain donors' decisions to give, we designed an online questionnaire referring to the benefits of giving, reputation, psychological benefits, and values. Since our study was concerned with contribution behavior in the context of green and sustainable projects, we also included a measure of environmental awareness. In addition, geographical proximity between project initiator and donor [42, 43] and individual sociodemographic characteristics [44] have been demonstrated to be relevant in the context of crowdfunding. Appendix B in the S1 Appendix presents an outline of the questionnaire, while Appendix C in the S1 Appendix provides a detailed description of the variables used in the empirical analysis.

### 2nd Part—Inducing happiness

Second, at the end of the online survey, the affective primes were introduced in a mindfulness exercise frame. We chose this framing because it is convincing in the social context of green giving and therefore, should reduce non-compliance. Following recent experimental findings in social psychology on the effectiveness of eliciting positive emotions [6], we exposed the participants to the affective primes of active autobiographical recall and visual stimuli as the most effective instruments.

We distinguished between the recall of positive past experiences and the act of sharing them with the host organization by instructing participants to write down these events. The recall stimuli were further compared with a visual stimulus that addressed more unconscious and less controlled processing. This design aimed at disentangling the effect of distinct priming techniques. Employing a between-subjects design, participants were randomly assigned to one of five treatment groups or the control group based on the saved browser cookie (ID). (1) Treatment group 1 received the task of writing down an emotional experience within the last year. "Before you decide, we are interested in understanding the experiences that make you happy or cheerful. They can be anything; for example, the birth of a child, a relative's marriage, or success in your profession. Please describe an event in the last year that made you happy." We interpreted the writing task as an act of emotional self-disclosure associated with diminishing the social distance between the recipient of the information and the sender. (2) Treatment group 2 was asked to recall an event that made them happy within the last year without writing it down. (3) Treatment group 3 received a priming picture. Though the literature provides inconsistent findings regarding whether a picture with happy looking people or a picture with sad looking people is more effective in generating donations, recent findings in social psychology suggest that the effectiveness of the picture strongly depends, amongst other things, on the psychological involvement with the charity [45]. We opted for a picture showing four smiling children in untouched nature as in our study context, it can be assumed that the study participants exhibit a high level of pro-social engagement, which the platform provider also confirmed in advance. (4) Treatment group 4 received the priming picture and the recall task. (5) Treatment group 5 received the priming picture and the writing task. The control group did not receive any treatment.

## 3rd Part—Incentivized dictator game

Third, the behavioral disposition to donate was measured by an incentivized dictator game that participants were asked to participate in at the end of the online survey. The game is based on an "All-or-Nothing" version of the dictator game where participants could not split the endowment [46]. Unlike most laboratory experiments, which address individuals as recipients of donations, we replace the individual recipient's position with two NGOs from which the dictator can choose. Respondents can use their participation fee of €5 to make one of the following three decisions once:

1. Give €5 from their earned money the green crowdfunding platform;

2. Keep the money and receive a personal voucher;

3. Give €5 from their earned money to another charity, namely an environmental organization.

Those participants who kept their money were offered a private voucher for a sustainable online shop. After making their choice, these participants were asked to note their e-mail address on a separate screen. To mitigate order effects, the three possible options appeared randomly.

Notably, while both charities offer a similar public good and are similar in terms of brand recognition, two differences have to be mentioned that are potentially relevant for the allocation decision: First, the crowdfunding platform was recognizable to the participants as the host of the survey, while the environmental organization was in no way connected. Second, while both NGOs offer a similar public good, namely environmental protection, the crowdfunding

platform primarily supports financing a broad range of sustainable projects, while the environmental organization emphasizes environmental activism.

## Results

### Sample description

Fig 1 presents participants' contributions in the dictator game by control and treatment groups for the set of variables in our analysis. The figure demonstrates that the proportion of donors does not vary significantly between treatments. Thus, at first glance, the treatments do not influence the decision to donate or not. However, they affect the allocative giving decision. Both the *Writing task* and the *Recall plus picture task* were the most effective primes for influencing allocation decisions in favor of the crowdfunding platform. Both treatments were also the most successful at inducing happiness and joy (see Table 2). The highest fraction of donations towards the environmental organization were reported in the "*Writing task plus picture*" treatment, followed by the "*Recalling task*" treatment.

Table 1 reports the summary statistics by control and treatment groups for treatment assignment and the set of individual characteristics as they appeared in our sample. Our data revealed that participants do not strongly perceive expectations from the social environment to contribute to sustainable projects. This is not surprising, as many of the donation-based crowdfunding sites are based on anonymous donations, suggesting that individuals' concern about their social image gain from donating is low. About a third, however, report observing crowdfunding behavior in their social environment. Also, seeking benefits by engaging in green crowdfunding is not widespread in the sample. Only 24% strongly or very strongly agreed with the statement that they support sustainable projects and startups because they are interested in the exchange-goods the projects offer for contributing. Rather, the participants seem to attach importance to the psychological benefits of giving and to individual values and attitudes. 82% of participants indicate that it makes them feel good to contribute to sustainable projects and startups. Further, the participants display profound feelings of responsibility. 72% feel strongly or very strongly responsible for contributing to a sustainable project and almost half state that they contribute to green crowdfunding to support their region.

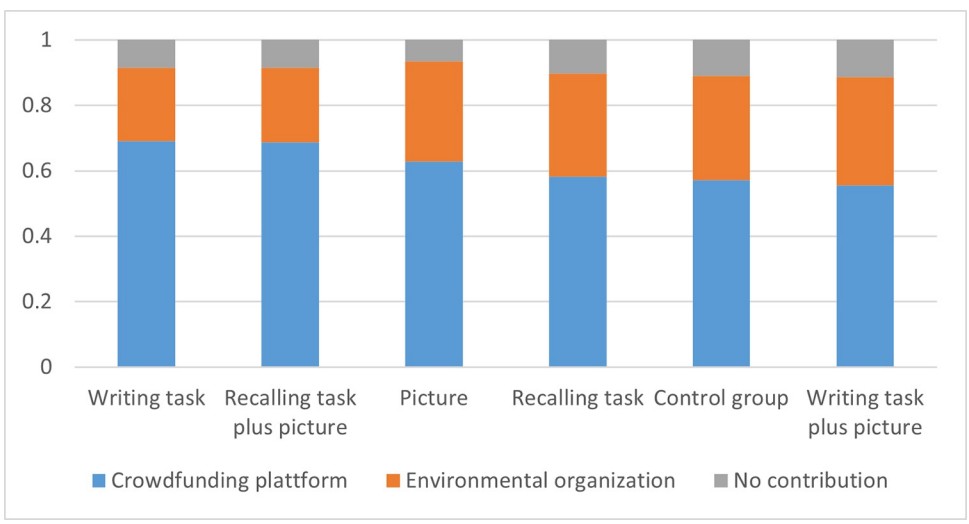

**Fig 1. Contributions by treatment (percent).**

**Table 1. Summary statistic of experimental design.**

| Treatment group | Control | Writing task | Recalling task | Picture | Recalling task plus picture | Writing task plus picture | Total |
|---|---|---|---|---|---|---|---|
| *Motivations related to reputational concerns* | | | | | | | |
| Expectations society (%) | 0.19 | 0.18 | 0.19 | 0.19 | 0.18 | 0.12 | 0.17 |
| Behavior family and friends (%) | 0.34 | 0.33 | 0.39 | 0.35 | 0.36 | 0.29 | 0.34 |
| *Benefits of giving* | | | | | | | |
| Reward seeking (%) | 0.26 | 0.22 | 0.2 | 0.23 | 0.24 | 0.28 | 0.24 |
| *Psychological benefits of giving* | | | | | | | |
| Warm glow feelings (%) | 0.88 | 0.81 | 0.83 | 0.81 | 0.77 | 0.82 | 0.82 |
| *Vales and attitudes related to giving* | | | | | | | |
| Feeling responsible (%) | 0.77 | 0.69 | 0.73 | 0.76 | 0.71 | 0.66 | 0.72 |
| Local identification (%) | 0.54 | 0.47 | 0.53 | 0.46 | 0.48 | 0.47 | 0.49 |
| Social trust (mean) | 5.54 | 5.55 | 5.61 | 5.79 | 5.7 | 5.34 | 5.59 |
| *Green values* | | | | | | | |
| Environmental concern (mean) | 4.82 | 4.91 | 4.96 | 4.88 | 5.09 | 4.86 | 4.92 |
| *Geographical proximity* | | | | | | | |
| Berlin (%) | 0.14 | 0.15 | 0.14 | 0.14 | 0.17 | 0.18 | 0.15 |
| *Socio-demographic characteristics* | | | | | | | |
| Age (mean) | 42.72 | 42.54 | 44 | 43.95 | 42.67 | 42.36 | 43 |
| Female (%) | 0.47 | 0.5 | 0.51 | 0.55 | 0.45 | 0.47 | 0.49 |
| Highest education (%) | 0.82 | 0.78 | 0.78 | 0.81 | 0.8 | 0.8 | 0.8 |
| Crowdfunding in 2017 (%) | 0.52 | 0.5 | 0.49 | 0.52 | 0.53 | 0.48 | 0.51 |
| *Number of observations* | 173 | 165 | 165 | 170 | 166 | 169 | 1008 |

An important striking characteristic of the participants in the online survey is that they exhibited strong environmental concerns. They reported an average value of 4.9 out of 6 on the NEP score. On the one hand, this posed a problem for our experiment since it suggested the disposition to donate in this group should already be high on average and difficult to stimulate further. On the other hand, it provided the chance to analyze our treatments' differential effects on the smaller subgroup of less green, less intrinsically motivated individuals. Furthermore, it is interesting to note that almost 15% of participants originated from Berlin, who, in other words, are a substantial subgroup with a local attachment to the crowdfunding platform.

Compared with the German Socio-Economic Panel, our sample is almost representative in terms of gender and age but unrepresentative regarding their social status because of our participants' high level of formal education. Given that the treatment groups seemed to be well-balanced in terms of observable characteristics, as displayed in Table 1, we concluded that the randomization process was successful.

## Manipulation check

To measure the effectiveness of inducing happiness, a parallel subject pool was used, which was not included in the primary analysis. Furthermore, our study adhered to the guidelines established by the crowdfunding platform, limiting the average duration of the study to 5 minutes per participant and the total study duration to 6 months. These restrictions limited our ability to explore various visual stimuli or more complex assessments of the emotional states of participants. The participants were provided with seven emotional states: three positive (happiness, joy, and surprise) and four negative emotional states (anger, annoyance, envy, and sadness). They were asked to rate the intensity of each emotion on an 11-point scale (from 0 = "not at all" to 10 = "very strong") (see [3]). Table 2 reports the results of Maximum Likelihood

**Table 2. Maximum Likelihood (ML) estimates in ordered probit models for the emotional intensity (happiness, joy, surprise, anger, annoyance, envy, sadness) by treatment group.**

|  | Happiness | Joy | Surprise | Anger | Annoyance | Envy | Sadness |
|---|---|---|---|---|---|---|---|
|  | (1) | (2) | (3) | (4) | (5) | (6) | (7) |
| *Treatments* (base: *control group*) |  |  |  |  |  |  |  |
| Writing task | 1.1729*** | 1.1193*** | 0.2499 | 0.0271 | 0.1242 | −0.0162 | 0.2452 |
|  | (4.95) | (4.86) | (1.00) | (0.09) | (0.38) | (−0.05) | (0.84) |
| Recalling task | 1.0312*** | 1.0357*** | 0.1413 | 0.1467 | 0.1815 | 0.2278 | 0.3870 |
|  | (4.44) | (4.54) | (0.59) | (0.52) | (0.58) | (0.76) | (1.35) |
| Picture | 0.3858* | 0.3803* | 0.3975 | 0.2372 | 0.2739 | 0.2196 | 0.1691 |
|  | (1.71) | (1.72) | (1.59) | (0.83) | (0.87) | (0.71) | (0.56) |
| Recalling task plus picture | 1.2856*** | 1.1936*** | 0.2148 | −0.1066 | −0.0511 | −0.0943 | 0.0593 |
|  | (4.96) | (4.74) | (0.78) | (−0.36) | (−0.16) | (−0.29) | (0.20) |
| Writing task plus picture | 0.9777*** | 1.0154*** | −0.0295 | 0.0671 | 0.0698 | 0.0030 | **0.5124*** |
|  | (4.14) | (4.32) | (−0.12) | (0.23) | (0.21) | (0.01) | (1.73) |

Note: The table presents the Maximum Likelihood (ML) estimates in ordered probit models for the emotional intensity by treatment group using 403 observations. Figures in parentheses are robust z-statistics. Coefficients that are statistically significant at the 1%, 5%, and 10% level are marked with ***, **, and *, respectively.

(ML) estimations of ordered probit models for the emotional intensity and the five treatment groups among the subsample of 403 respondents.

The first two columns reveal the relevance of the treatments for the induction of happiness or joy. As expected, all treatments are significantly positively correlated with increasing values of happiness or joy, and thus elicited these positive emotions in the participants. The highest levels of happiness (joy) were reported in the "*Recalling task plus picture*" treatment, followed by the "*Writing task*" treatment. Column five also shows that the writing task plus the picture led to a statistically significant emotional response, but interestingly also to sadness. We can conclude that the priming techniques used to induce emotions made the participants happier and gave them joy. Furthermore, it must be noted that the combination of the writing task with the picture seemed to induce unintended negative incidental emotions. The combination of the different stimuli thus had nonadditive effects on the emotional state of participants.

## Basic estimation results

To explore formally the effects of positive emotions on the allocation of online donations, we first estimated pooled intention-to-treat (ITT) effects. Table 3 demonstrates estimates applying the multinomial probit model with the allocative giving decision (measured by the variable "*Contribution behavior*" capturing the mutually exclusive alternatives: *no contribution*,

**Table 3. Average discrete probability effect of positive emotions on contribution behavior.**

|  | No contribution | Contribution to the crowdfunding platform | Contribution to the environmental organization |
|---|---|---|---|
| *Treatments* (base: *control group*) |  |  |  |
| Positive emotion | −0.020 | 0.0565 | −0.0365 |
|  | (−0.78) | (1.37) | (−0.94) |

Note: The table reports the average discrete probability effect of positive emotions on contribution behavior in the multinomial probit model using 1,008 observations. The dependent variable "Contribution behavior" includes the mutually exclusive alternatives: no contribution, contribution to the crowdfunding platform (base category), contribution to the environmental organization. Figures in parentheses are robust z-statistics. Coefficients that are statistically significant at 1% (5%, 10%) level are marked with *** (**, *), respectively.

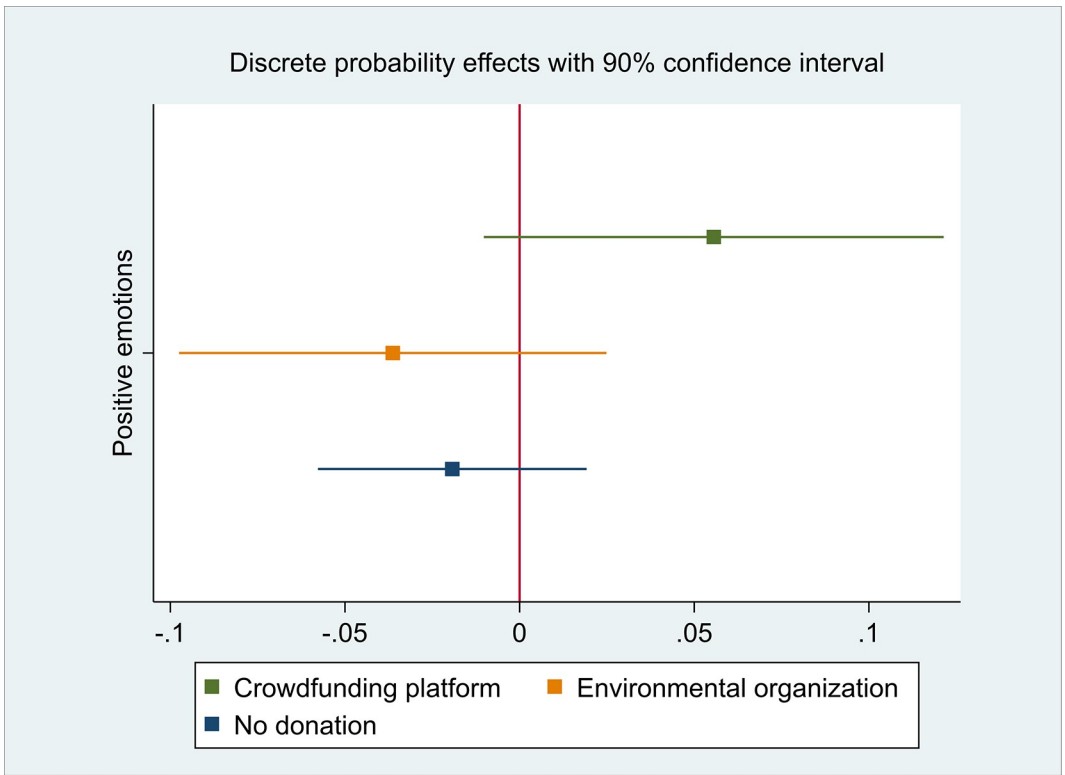

**Fig 2. 90% confidence intervals for the average discrete probability effect of positive emotions.**

*contribution to the crowdfunding platform* (base category), *contribution to the environmental organization*) as dependent variable. Considering only the respondents for whom the treatments were successfully assigned, estimates of local average treatment effects (LATE) show statistically significant effects of the recalling task (see Appendix E in the S1 Appendix).

Results in Table 3 do not confirm our hypothesis that inducing positive emotions positively affects donation behavior. We found neither statistically significant results on the effect of happiness on the contribution behavior. We present the 90% confidence interval (CI) around the estimated main treatment effect in the multinomial probit model in Fig 2. From an ex-post perspective, we cannot rule out a small decrease of 1.1 percentage points, but we cannot rule out that treated participants show a 12.4 percentage points increased probability for choosing to donate to the crowdfunding platform compared to participants in the control group who also choose to donate to the crowdfunding platform either if we maximally accept a tolerance of 10% for a Type I error.

Previous results in the literature have been mixed in this regard. The majority of studies found positive effects, while a small number of studies found null results. While being in line with Ibanez et al. (2017), who found emotional induction does not have any impact on the probability of giving in a dictator game with an environmental NGO as recipient, this result contrasts with the bulk of laboratory evidence on the effectiveness of inducing happiness for generosity [1, 3, 21–23]. One further possible explanation, might be "ceiling" effect, i.e., the visitors of the crowdfunding platform are a highly selective sample of supporters of this platform or pro-social/pro-environmental causes. Their presumably strong pro-social commitment might leave little room for any potential positive treatment effects of the affective primes.

### Estimation results by treatment

Next, we examine the effects of social process-driven, cognitive, and visual affective primes on the allocation decision. Table 4 presents discrete effects by treatment of a multinomial probit regression on the probability of contributing. The results from the regression analyses support Hypothesis 2. While none of the treatments were more or less effective in stimulating donations, as the first column of Table 4 reveals, inducing affective primes that promote the sharing of positive emotions with the crowdfunding platform, in comparison with the control condition, affected the allocation of donations, as the second column of Table 4 demonstrates. Respondents who were invited to write down a happy event from the past year and share it with the crowdfunding platform exhibited a significantly higher probability of donating to the host organization. Participants who received this treatment exhibited an 11.9 percentage point increase in the probability of choosing to contribute to the host platform compared to participants in the control group. Additionally, they showed a 9.37 percentage point decrease in the estimated probability of choosing to donate to the environmental organization compared to the control group.

The less social process-driven cognitive stimuli of the recall task revealed no statistically significant effect on the contribution behavior; neither did the more immediate priming technique of inducing positive emotions by visual stimuli. While in this case, the manipulation of the participants' emotional state was only weak (see manipulation check), the result complements previous contributions showing that pictures (as affective primes) can be used to trigger donations in the dictator game. Small and Verrochi (2009) [19], for instance, report that a picture with sad-looking children generated more donations than did a picture with happy looking children. They, however, did not demonstrate differences between the happy and neutral expression conditions.

While neither the recall task nor the picture alone were effective in stimulating online giving, the combination of the two did so. Those participants who were asked to recall a happy autobiographical event and were shown a picture simultaneously displayed a higher probability of contributing to the crowdfunding platform than participants in the control group. The recall task plus picture increased the choice probability for crowdfunding by 11.45 percentage

**Table 4. Average discrete probability effects by treatment group for contribution behavior.**

|  | No contribution | Contribution to the crowdfunding platform | Contribution to the environmental organization |
|---|---|---|---|
| *Treatments (base: control group)* |  |  |  |
| Writing task | −0.0250 | 0.1187** | −0.0937* |
|  | (−0.78) | (2.28) | (−1.95) |
| Recalling task | −0.0068 | 0.0096 | −0.0028 |
|  | (−0.20) | (0.18) | (−0.05) |
| Picture | −0.0451 | 0.0572 | −0.0120 |
|  | (−1.49) | (1.08) | (−0.24) |
| Recalling task plus picture | −0.0255 | 0.1145** | −0.0890* |
|  | (−0.79) | (2.20) | (−1.85) |
| Writing task plus picture | 0.0026 | −0.0160 | 0.0134 |
|  | (0.08) | (−0.30) | (0.27) |

Note: The table reports the average discrete probability effects by treatment group in the multinomial probit model using 1,008 observations. The dependent variable "Contribution behavior" includes the mutually exclusive alternatives: no contribution, contribution to the crowdfunding platform (base category), contribution to the environmental organization). Figures in parentheses are robust z-statistics. Coefficients that are statistically significant at 1% (5%, 10%) level are marked with *** (**, *), respectively.

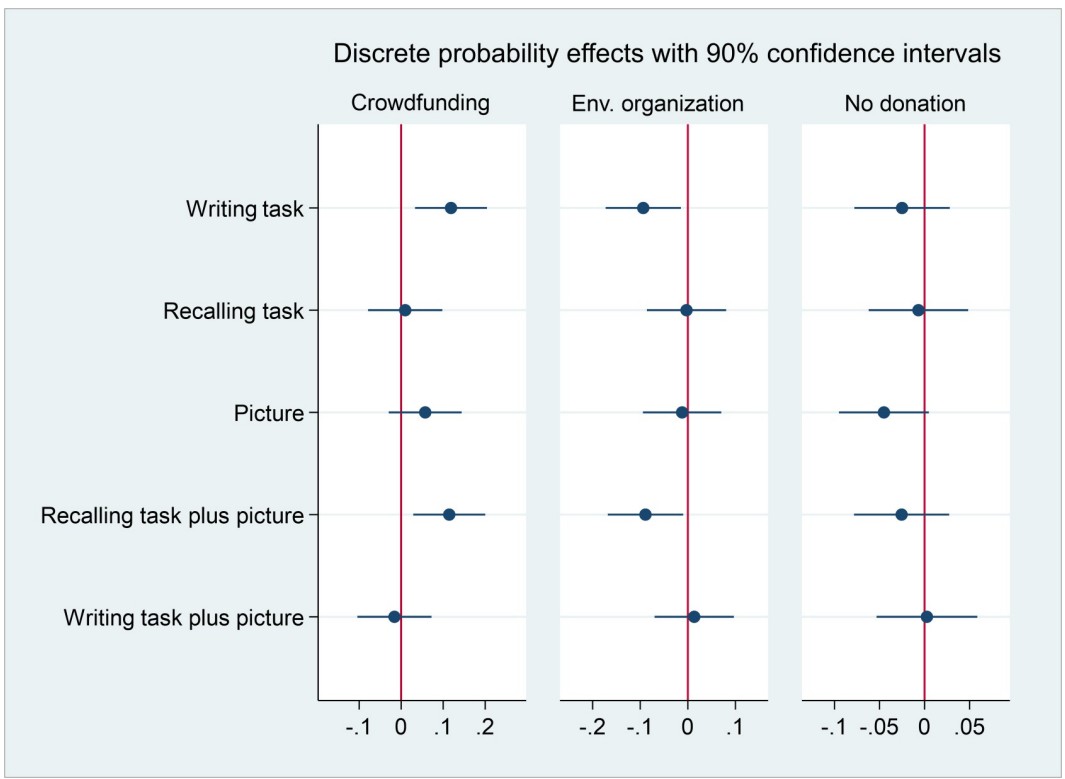

**Fig 3. 90% confidence intervals for the average discrete probability effects by treatment group for contribution behavior.**

points, whereas it decreased the choice probability for the environmental organization by 8.9 percentage points, compared to the control treatment. This result could be due to the recall task being less effective in inducing happiness and joy compared with the combination of the recall task with the picture.

However, the combination of the writing task and picture did not yield statistically significant effects. This points to the less theoretical and more technical insight that the combination of autobiographical writing tasks with visual stimuli is not additive (see above). We mainly attributed this to the incidental sadness induced by the stimulus (see Table 1), which could impair the reduction in social distance felt by disclosing the positive emotion. Table D.1 in the Appendix D of S1 Appendix demonstrates that the results are robust for several confounders.

We present the 90% CI around the estimated treatment effects in the multinomial probit model in Fig 3. A value lower than -0.079 or higher than 0.098 should have been observed to reveal an effect of the recalling task on the probability of donating to the crowdfunding platform. Although we cannot rule out zero, we also cannot rule out an 9.8 percentage points increased choice probability for donations to the host charity either if we maximally accept a tolerance of 10% for a Type I error.

Finally, we conduct an analysis to assess the robustness of our results to adjustments for multiple hypotheses testing. For this purpose, we assume that Hypothesis 2 ("The crowding out effect") is represented by the five parameters estimated in column 2 of Table 4. In this case, the results maintain significance at the 10% level after adjusting for the False Discovery Rate (FDR), but not after applying the Bonferroni Correction, which is a more conservative correction method for multiple comparisons.

## Heterogeneity analysis

Analyzing heterogeneous treatment effects is of particular importance in a field-experimental setting such as ours, where many participants are probably highly motivated a priori and already exhibit strong environmental concern. To disentangle the individual responsiveness to the type of affective prime applied, we examined the interaction between the effects of the treatments and the level of environmental concern opting for the split sample approach.

The sample was divided into two subsamples based on different values of the variable "*Environmental concern high*" allowing us to investigate how the induced affective primes operate for participants with varying levels of environmental concerns. We compare the estimated average discrete probability effects of treatments for individuals with below- and above-median environmental concern. Table 5 displays the average discrete probability effects by treatment group at different levels of "Environmental concern high" for contribution behavior. The results in Table 5 indicate no statistically significant changes in the choice probability for contributing in general (column 1), to the crowdfunding platform (column 4), or to the environmental organization (column 7) among respondents with environmental concerns below the sample median. However, we found effects of the treatments on the contribution behavior of respondents whose environmental concern was above the median of the sample.

For participants who expressed above-median levels of environmental concern, treatments one, three, and four led to a significant increase in donations to the crowdfunding platform, as compared to those in the control group who had similar levels of intrinsic motivation. For example, the fourth column illustrates that respondents who completed the writing task were 18.19 percentage points more likely to contribute to the crowdfunding platform than those in

**Table 5. Average discrete probability effects by treatment group at different values of "Environmental concern high" for contribution behavior.**

| | No contribution | | | Contribution to the crowdfunding platform | | | Contribution to the environmental organization | | |
|---|---|---|---|---|---|---|---|---|---|
| | Below median concern | Above median concern | Difference | Below median concern | Above median concern | Difference | Below median concern | Above median concern | Difference |
| | (1) | (2) | (3) | (4) | (5) | (6) | (7) | (8) | (9) |
| *Treatments (base: control group)* | | | | | | | | | |
| Writing task | 0.0138 | -0.0389 | 0.0527 | -0.0175 | 0.1819*** | 0.1995* | 0.0038 | -0.1430** | -0.1468 |
| | (0.21) | (-1.04) | (0.71) | (-0.18) | (2.93) | (1.75) | (0.05) | (-2.44) | (-1.44) |
| Recalling task | 0.0988 | -0.0517 | 0.1505* | -0.072 | 0.0517 | 0.1237 | -0.0269 | 0.000 | 0.029 |
| | (1.40) | (-1.41) | (1.89) | (-0.77) | (0.79) | (1.09) | (-0.35) | (0.00) | (0.27) |
| Picture | -0.0011 | -0.0629* | 0.0618 | -0.08 | 0.1221* | 0.2021* | 0.0811 | -0.0592 | -0.1404 |
| | (-0.02) | (-1.78) | (0.89) | (-0.85) | (1.92) | (1.78) | (0.95) | (-0.96) | (-1.34) |
| Recalling task plus picture | -0.0115 | -0.03 | 0.0185 | 0.1283 | 0.1395** | 0.0112 | -0.1168 | -0.1095* | 0.0073 |
| | (-0.18) | (-0.79) | (0.24) | (1.38) | (2.25) | (0.10) | (-1.56) | (-1.86) | (0.08) |
| Writing task plus picture | 0.1023 | -0.0431 | 0.1454* | -0.0427 | 0.00 | 0.0427 | -0.0596 | 0.0431 | 0.1027 |
| | (1.48) | (-1.15) | (1.85) | (-0.47) | (-0.00) | (0.38) | (-0.81) | (0.67) | (1.06) |
| Number of observations | 281 | 727 | | 281 | 727 | | 281 | 727 | |

Note: The table reports the average discrete probability effects by treatment group at different values of "Environmental concern high" in the multinomial probit model using 1,008 observations. The dependent variable "Contribution behavior" includes the mutually exclusive alternatives: no contribution, contribution to the crowdfunding platform (base category), contribution to the environmental organization). Figures in parentheses are robust z-statistics. Coefficients that are statistically significant at 1% (5%, 10%) level are marked with *** (**, *), respectively

the control group with an equal level of environmental concern (p = 0.003). Vice versa, treatments one and four resulted in a statistically significant reduction in donations to the environmental organization among participants with above-median levels of environmental concern, as compared to individuals in the control group with similarly high levels of environmental concern. For example, the results from the eighth column show that respondents who completed the writing task were 14.30 percentage points less likely to contribute to the environmental organization than those in the control group with an equal level of environmental concern.

Regarding the differences between individuals ranking below- and above-median in environmental concern, the third column reveals differences in how inducing emotions interacted with environmental concern for the level of total contributions. Our findings indicate that the average change in the predicted probability of donating money in the dictator game when asked to recall a positive event in the past differed between by 15.05 percentage points. A similar pattern was apparent for the writing task plus picture. Specifically, respondents exhibiting high environmental concern displayed a higher marginal probability effect of this treatment on their total contribution behavior. In addition, the sixth column reveals that the average change in probability of donating to the crowdfunding platform for respondents with above-median environmental concern who shared a happy event with the crowdfunding platform exhibit an increase of almost 19.95 percentage points compared with those who reported below-median environmental concern. The average discrete probability effect of the visual stimuli, i.e. the picture, was 20.21 percentage points higher for people ranking above-median environmental concerns compared with those ranking below. However, we should note that none of these exploratory results retain statistical significance after correcting for multiple hypotheses testing using either the False Discovery Rate (FDR) or the Bonferroni Correction adjustment.

## Discussion

Do positive emotions generate donations, or do they create spillover effects that benefit adjacent organizations? Laboratory findings on the positive effect of emotions on giving behavior are not confirmed in an online field setting. Positive incidental emotions do not exhibit a positive effect on giving behavior per se. We provide null results on giving versus non-giving. We found no statistically significant effects of the affective primes on the decision to donate or not. Based on our data we do not confirm Hypothesis 1. In this context, it must be noted that the overall willingness to donate in the sample was extremely high, comparable to the finding of Bekkers (2007). He finds that the overwhelming majority of participants in the study (94.3%) keep the reward earned for participation. Demonstrating, the effects of inducing and communicating positive emotions may be more viable in a lab situation, where more variance in the a-priori disposition to donate can be secured by design. This, however, has the downside that it might produce results with little applicability in field situations where the overall disposition to donate is rather high (as in our case) or extremely low (like door-to-door fundraising).

We find, however, some evidence that positive emotions unrelated to the decision to donate have an effect on the allocation of donations. Affective primes that aimed at establishing a social bond between donor and recipient (writing task) and a combination of primes that were designed to initiate positive cognitive processes (recall task plus picture) were the most effective at stimulating allocation decisions in favor of the targeted crowdfunding platform in our study. In other words, our findings do not indicate the existence of spillover effects that benefit adjacent organizations. Conversely, priming techniques that aim at establishing a social bond between donor and recipient, as well as primes that initiate positive cognitive processes, are

worth a try as behavioral tools to capture a larger share of donations in a multi-charity context. However, it is important to note that while these results are robust to a False Discovery Rate adjustment for multiple hypotheses testing, they do not withstand the more conservative Bonferroni Correction. As a result, we express moderate confidence in our support for Hypothesis 2 ("Crowding out effect"). Moreover, we found no empirical evidence for a substantial effect of the recall task in our data. This caveat indicates that the combination of autobiographical memory tasks and visual stimuli is not additive and merits further research. The data further reveal no statically significant results for the combination of writing tasks with the picture. As combining the writing task with the picture stimulus induced both positive and negative emotions, this empirical result is difficult to interpret. This also points to a first limitation of our work: The role that incidental negative emotions play in stimulating or inhibiting donations by disclosing positive emotions needs to be further disentangled by further research.

Our explorative results also indicate tentatively that evoking positive emotions triggers contribution behavior to the host charity that provides the emotional stimulus even when the potential donor base is already highly intrinsically motivated. Controlling for heterogeneous treatment effects depending on the degree of intrinsic motivation, we found a statistically significant effect of happiness on the donation decision primarily in participants with high environmental concerns. As a caveat, these results are not robust to multiple hypothesis testing. However, they fit with recent findings on priming effectiveness depending on the psychological involvement with the charity [45]. A strong psychological involvement with a green cause is a reasonable assumption among participants with high environmental concerns. Therefore, our study provides some, albeit weak, evidence that emotional priming might be a viable option for stimulating donations in a target audience that exhibits high intrinsic motivation levels to donate. The dominance of a highly intrinsically motivated audience, as revealed in our online field experiment, should be fairly common in practical application contexts of online fundraising campaigns and is thus non-neglectable. Most charities will encounter a strong homogeneity among the members of the public they can effectively reach for fundraising purposes. Our conclusion should thus be informative for charities that operate online campaigns to raise funds.

The paper in question explores the impact of emotions on pro-social behavior, specifically charitable giving, a topic that has already been acknowledged as important in the existing literature. Past studies have established a relationship between positive emotions, particularly happiness, and increased generosity. This paper contributes to that understanding by testing the effect of affective primes as behavioral tools in an online field setting, an approach not frequently used in the existing literature. While previous research has generally focused on the impact of emotions experienced privately by individuals, this paper broadens the scope to include the social-regulatory functions of expressed emotions. This unique perspective focuses on the impact of shared feelings and communication about one's emotions in establishing and deepening positive social bonds, a factor that can significantly enhance donations. This study also contributes to the existing literature by examining the role of emotions in a multi-charity context, an area that hasn't been sufficiently addressed. Existing research indicates that evoking emotions can increase donations to a single charity, but the implications in a context where multiple charities are competing for donations are not clear. This paper attempts to fill this gap by examining whether certain fundraising strategies, specifically those aimed at eliciting incidental emotion, increase donations to the host charity or divert them to another charity.

However, our study has several limitations. It reveals that online lab-in-field experiments suffer from strong homogeneity of the target audience of online platforms. The self-selection of participating in the experiment amplifies this limitation, as it can introduce correlations

between causally unrelated background variables that co-determine a participant's susceptibility to positive emotions and willingness to donate. This effect could exacerbate ceiling effects that lead to null results, as observed in the present case. In principle, this problem can be mitigated by another feature of online experiments: the low or zero marginal costs of increasing the sample size [47]. In practice, however, this is not as simple as in theory because limitations exist in the number of potential participants: green crowdfunding platforms are still, for the most part, niche services, and they do not attract millions of visitors.

A further limitation of our research resulted from the limited attention charities can draw to academic experiments. In practice, we were strongly limited in terms of the average duration of our experiment per individual. This entailed compromises in the precision of the measurement instruments of our experiment. It also limited the assessment of the manipulations to a very simple post-hoc between-groups check. Therefore, some crucial causal connections that we assume to be at work in our experiment are only assumed and not demonstrated directly. For instance, the effect of the increase in the level of positive emotion on individual donation behavior was always implied but not itself measured.

The finding that incidental positive emotions potentially affect the allocation of donations behavior suggests that organizations seeking to increase charitable giving could benefit from incorporating activities or prompts that evoke positive emotions into their community-outreach processes. Charities can create opportunities for potential donors to get involved with the charity and feel a sense of closeness. This could include organizing volunteer events or social activities that bring people together and help them feel connected to the organization. For example, charities could create events that are enjoyable and fun, such as charity runs or auctions, to evoke positive emotions and create a sense of community among potential donors. Additionally, charities can offer social support to potential donors by providing resources or information that help them feel more connected to their cause. For example, offering groups or online forums for people interested in a particular issue can create a sense of familiarity and connection. Furthermore, practitioners could consider using digital media to engage with potential donors and create opportunities for them to share positive emotional experiences. Such strategies could help potential donors feel more connected to the charity and more likely to support the cause there than elsewhere.

Future research in the field of emotions and pro-social behavior holds significant potential for generating valuable insights by exploring several key directions. Firstly, conducting cross-cultural comparisons would provide a deeper understanding of how emotions and pro-social behavior vary across different cultures. Investigating the influence of cultural values, norms, and societal expectations on the interplay between emotions and pro-social behaviors can facilitate the development of more effective, culture-specific interventions to promote pro-social behavior. Furthermore, studying the role of peer influence is crucial as peers have a profound impact on individuals' attitudes, beliefs, and behaviors. By examining emotional contagion, where the emotions expressed by peers influence an individual's emotional state, researchers can elucidate how peers shape pro-social behavior. Establishing whether positive emotions displayed by peers, such as happiness or gratitude in response to pro-social actions, can establish social norms that promote pro-social behavior is essential. The emotional connection and validation individuals receive from their peers can strengthen their motivation to engage in pro-social behavior. While existing research predominantly focuses on the influence of positive emotions, further exploration of the role of negative emotions in pro-social behavior is warranted. Investigating how sharing negative emotions can decrease the social distance between donors and recipient organizations can offer compelling insights. Understanding the potential of negative emotions to motivate pro-social behavior provides a comprehensive understanding of the range of emotional drivers behind pro-social actions. Lastly, investigating

the influence of individuals' emotion regulation abilities on their pro-social behavior can provide illuminating insights. Analyzing whether individuals who possess effective emotion regulation strategies are more or less likely to engage in pro-social behavior sheds light on the underlying mechanisms. This line of research deepens our understanding of the emotional processes involved in pro-social behavior and contributes to the development of interventions that foster effective emotion regulation strategies to promote pro-sociality.

## Conclusion

Our study provides evidence that incorporating activities or prompts that evoke positive incidental emotions can potentially affect donation behavior and benefit organizations seeking to increase charitable giving in a multi charity context. However, our findings provide no evidence that positive incidental emotions directly influence the decision to give or not. Nonetheless, we provide useful insights for charities operating online fundraising campaigns, especially if they face a highly intrinsically motivated audience, as in the case of the participants of our study. In particular, we find that strategies that produce positive incidental emotions by reducing the social distance to the charity might be worth a try to attract a bigger share of donations. Finally, it is important to acknowledge the limitations of our research, including the homogeneity of the online platform's target audience, the self-selection bias of participants, and the constraints on experimental precision and manipulations. Due to the "lab-in-field" setting we could only focus on positive emotions. Further field research should address the role of incidental negative emotions and explore the causal connections between positive emotions and donation behavior in larger samples.

## Supporting information

**S1 Appendix.**
(DOCX)

## Author Contributions

**Conceptualization:** Anja Köbrich Leon, Janosch Schobin.

**Data curation:** Anja Köbrich Leon, Janosch Schobin.

**Formal analysis:** Anja Köbrich Leon, Janosch Schobin.

**Investigation:** Anja Köbrich Leon, Janosch Schobin.

**Methodology:** Anja Köbrich Leon, Janosch Schobin.

**Project administration:** Anja Köbrich Leon, Janosch Schobin.

**Resources:** Anja Köbrich Leon, Janosch Schobin.

**Writing – original draft:** Anja Köbrich Leon, Janosch Schobin.

**Writing – review & editing:** Anja Köbrich Leon, Janosch Schobin.

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
