## [Decision Letter · Decision Letter 0]

11 Jan 2023

PONE-D-22-32907Get the happiness out – An online experiment on the causal effects of positive emotions on givingPLOS ONE

Dear Dr. Anja Köbrich Leon,

Thank you for submitting your manuscript to PLOS ONE. After careful consideration, we feel that it has merit but does not fully meet PLOS ONE’s publication criteria as it currently stands. Therefore, we invite you to submit a revised version of the manuscript that addresses the points raised during the review process. We have heard from two reviewers; one recommends a minor revision, and the other one a major revision. Both have raised concerns that you can address in your revision. After reading the paper myself, I think you can improve the paper by following the reviewers' suggestions. If you refrain from implementing specific requests, please explain why you do so.

Even though the paper's status is "major revision required," and I cannot guarantee anything, it is also true that I can foresee a straightforward path to acceptance.

We look forward to receiving your revised manuscript.

Kind regards,

Jaume Garcia-Segarra

Academic Editor

PLOS ONE

Journal Requirements:

2.Please provide additional details regarding participant consent. In the Methods section, please ensure that you have specified (1) whether consent was informed and (2) what type you obtained (for instance, written or verbal). If your study included minors, state whether you obtained consent from parents or guardians. If the need for consent was waived by the ethics committee, please include this information.

3. Please ensure that you include a title page within your main document. We do appreciate that you have a title page document uploaded as a separate file, however, as per our author guidelines (http://journals.plos.org/plosone/s/submission-guidelines#loc-title-page) we do require this to be part of the manuscript file itself and not uploaded separately.

Reviewers' comments:

Reviewer's Responses to Questions

**Comments to the Author**

1. Is the manuscript technically sound, and do the data support the conclusions?

Reviewer #1: Yes

Reviewer #2: Partly

2. Has the statistical analysis been performed appropriately and rigorously? 

Reviewer #1: Yes

Reviewer #2: Yes

3. Have the authors made all data underlying the findings in their manuscript fully available?

Reviewer #1: Yes

Reviewer #2: No

4. Is the manuscript presented in an intelligible fashion and written in standard English?

Reviewer #1: Yes

Reviewer #2: Yes

5. Review Comments to the Author

Reviewer #1: Dear authors,

When I started reading the abstract and the article I was turned off by the unclarity of the language and numerous mistakes (see examples below). Luckily, this seems to be a problem only at the beginning of the paper as the later part is much more clear and polished. So overall, after I finished reading the article, I think that it is interesting and deserves a publication.

However, I do have one pount regarding the design that I believe should be discussed. I wonder why the authors first announced the donation possibility and then introduced the stimuli. As this this leaved people more time between the announcement of donation possibility and the actual donation, this might have led people to use some kind of a self-control mechanism. As documented in (Exley & Petrie, 2016, 2018) leaving people time to deliberate leads people to look for excuses not to donate. To sum up: the stimuli should have been placed before the announcement of the donation decision.

Regarding the results by treatment, I do not really buy the crowding out effect. I guess the results do not survive MHT?

Heterogeneity: I am not especially convinced by the way the heterogeneity analysis is presented. I think you should just repeat your main specifications for the separate samples. If you have convergence problems, maybe simple linear estimations (in all Tables) would be a good alternative. The way you change the estimation method and the outcome variable it is difficult to compare the main estimation to the heterogeneity analysis.

Here are some comments on writing at the beginning of the article:

“Anticipated and experienced emotions related to the decision context are a driving

force for pro-social behavior.” Really? Where does this statement come from? Economists define other motives as driving force for charitable giving…

“dispositional affect” is a persistent trait and it does not square with induced emotions.

Please rewrite: “We provide field experimental evidence on causal effects of affective

primes, which aim to strengthen the social bond between donor and recipient, and

primes that aim to trigger positive cognitive processes are particularly useful in raising

donations to the target charity.”

“These effects tend depend on individual characteristics.”

“Second, charities eliciting positive incidental emotions without worrying about spillover to other charities.”

p. 20 (1) instead of (!)

Other related papers: (Wichman & Chan, 2022)

References:

Exley, C., & Petrie, R. (2016). Finding Excuses to Decline the Ask: A Field Experiment. 1–30. https://doi.org/10.2139/ssrn.2743207

Exley, C., & Petrie, R. (2018). The impact of a surprise donation ask. Journal of Public Economics, 158, 152–167. https://doi.org/10.1016/J.JPUBECO.2017.12.015

Wichman, C. J., & Chan, N. W. (2022). Preheating prosocial behavior.

Reviewer #2: Thank you for the opportunity to review this manuscript. I have experience both doing experimental research as well as reviewing many experimental studies. Overall, I find this study rigorous and its design and analysis. I do have some questions, largely from a clarity standpoint, to improve it's contribution to the literature.

1) Theory.

*The introduction discusses the theories of interest for this study, but without definition. To me, not being familiar with this area of psychology research, the intro was largely unintelligible to me, and I had to read the longer lit review more carefully to understand your framework.

*It seems to me that one of the big questions underpinning this line of inquiry is the causal direction - are happier people more likely to give, or does giving make you happy? I think you are trying to get at the former, by eliciting/reminding people of feelings of happiness before asking them to give. However, it still remains to be seen if those who did share a happy memory were more likely to give - can you qualitatively review the statements from the written task? But then, does having an easily recalled happy memory (does that actually make you happier?) make you more likely to give?

*I'm unsure about the option to give to the other environmental group, theoretically. What is that testing, or not testing?

2) Method.

*I couldn't really picture the setting/design of this experiment, and I read the appendix. Are these people who were coming to the crowdfunding platform to make a contribution to one of the asks there (and thus already primed to give? - this may explain why they were much more likely to give over Bekkers, 2017). Were they then asked to complete this survey before they were able to access the crowdfunding projects? Can you clarify this?

*I'm not sure what this as a "field experiment" over a standard "survey (lab style) experiment". There are pros and cons with each. Perhaps clarify that?

*I have gotten used to seeing a graphic showing the treatment assignments in these types of papers (i.e. what happens when people enter the experiment).

*Did you give people the opport8unity to opt out of the survey and continue to the crowdfunding platform? If so, can you describe the potential impact of selection bias in this situation?

*Were the informed of the 5 Euro prior to starting the experiment? I'm not sure how that 'mitigates' the "experimenter demand effects". (p. 4).

3) Implications.

*The findings for this study seem important for understanding the various psychological mechanisms to incentivize volunteering, but I'd like to see some discussion about what the practical implications might be for practitioners. Asking donors to engage in some activity prior to giving may help spur these attitudes, but could also lead many, many donors to stop the process of completing a donation. CAn you

6. PLOS authors have the option to publish the peer review history of their article (what does this mean?). If published, this will include your full peer review and any attached files.

Reviewer #1: No

Reviewer #2: No

---

## [Author Response · Author response to Decision Letter 0]

4 Apr 2023

Thank you very much for taking the time to give us guidance on improving our study “Get the happiness out – An online experiment on the causal effects of positive emotions on giving”. 

We value your thoughtful comments highly. We tried to incorporate them into the text and to respond to them as fully as possible. Please find our detailed responses to your suggestions below. You will find our direct responses to each suggestion (in italics) below. We worked through all of the comments and remarks, and we think that our paper has improved significantly as a result of us following them. We have revised our paper substantially: some parts have been rewritten or supplemented, and others have been dropped. 

A detailed, point-by-point response to the raised questions can be found in the files titled "Response to Reviewer."

---

## [Decision Letter · Decision Letter 1]

16 May 2023

PONE-D-22-32907R1Get the happiness out – An online experiment on the causal effects of positive emotions on givingPLOS ONE

Dear Dr. Anja Köbrich Leon,

Thank you for submitting your manuscript to PLOS ONE. After careful consideration, we feel that it has merit but does not fully meet PLOS ONE’s publication criteria as it currently stands. Therefore, we invite you to submit a revised version of the manuscript that addresses the points raised during the review process. We have received a report on your manuscript. The reviewer is satisfied with your improvement. There is only a suggestion for reorganizing the sections. Thus, consider this letter as a conditional acceptance of your paper. Please, answer the request of the reviewer. If you refrain from implementing the changes, explain what reasons drive your decision. The manuscript will not come back to the referees.

We look forward to receiving your revised manuscript.

Kind regards,

Jaume Garcia-Segarra

Academic Editor

PLOS ONE

Journal Requirements:

Reviewers' comments:

Reviewer's Responses to Questions

**Comments to the Author**

1. If the authors have adequately addressed your comments raised in a previous round of review and you feel that this manuscript is now acceptable for publication, you may indicate that here to bypass the “Comments to the Author” section, enter your conflict of interest statement in the “Confidential to Editor” section, and submit your "Accept" recommendation.

Reviewer #2: All comments have been addressed

2. Is the manuscript technically sound, and do the data support the conclusions?

Reviewer #2: Yes

3. Has the statistical analysis been performed appropriately and rigorously? 

Reviewer #2: Yes

4. Have the authors made all data underlying the findings in their manuscript fully available?

Reviewer #2: (No Response)

5. Is the manuscript presented in an intelligible fashion and written in standard English?

Reviewer #2: Yes

6. Review Comments to the Author

Reviewer #2: Thank you for the opportunity to review this revision. The paper is in good shape now. However, the order of sections seemed odd - limitations the last paragraph, hypotheses after research design. I would structure the paper 1) Intro 2) Literature Review 3) Hypotheses 4) Research Design 5) Findings - Descriptive Statistics, then Analysis 6) Discussion 7) Conclusion

Don't forget to really focus on the contribution your paper makes to the literature.

7. PLOS authors have the option to publish the peer review history of their article (what does this mean?). If published, this will include your full peer review and any attached files.

Reviewer #2: No

---

## [Author Response · Author response to Decision Letter 1]

3 Jul 2023

Dear Mr. Garcia-Segarra, dear Referee

Thank you again very much for taking the time to give us guidance on improving our paper “Get the happiness out – An online experiment on the causal effects of positive emotions on giving”. We value your and the reviewer’s thoughtful comments highly. 

In response to the reviewer's suggestions, we have made the necessary adjustments to address his concerns regarding the structure of the paper. Furthermore, we have conscientiously revised the discussion and conclusion sections, ensuring meticulous alignment with the prescribed criteria set forth by PlosOne. Additionally, we have explicitly addressed our contributions within these sections.

We greatly appreciate your valuable input and continued support.

Sincerely,

---

## [Decision Letter · Decision Letter 2]

7 Aug 2023

Get the happiness out – An online experiment on the causal effects of positive emotions on giving

PONE-D-22-32907R2

Dear Dr. Köbrich-Leon,

We’re pleased to inform you that your manuscript has been judged scientifically suitable for publication and will be formally accepted for publication once it meets all outstanding technical requirements.

Kind regards,

Jaume Garcia-Segarra

Academic Editor

PLOS ONE

Additional Editor Comments (optional):

Reviewers' comments:

Reviewer's Responses to Questions

**Comments to the Author**

1. If the authors have adequately addressed your comments raised in a previous round of review and you feel that this manuscript is now acceptable for publication, you may indicate that here to bypass the “Comments to the Author” section, enter your conflict of interest statement in the “Confidential to Editor” section, and submit your "Accept" recommendation.

Reviewer #2: All comments have been addressed

2. Is the manuscript technically sound, and do the data support the conclusions?

Reviewer #2: Yes

3. Has the statistical analysis been performed appropriately and rigorously? 

Reviewer #2: Yes

4. Have the authors made all data underlying the findings in their manuscript fully available?

Reviewer #2: Yes

5. Is the manuscript presented in an intelligible fashion and written in standard English?

Reviewer #2: Yes

6. Review Comments to the Author

Reviewer #2: This paper is clearly laid out and written, and was easy to follow. I think the revisions have brought it a long way.

It seems to me that a lot of the additional analysis was provided to make sure the authors had something to say even in the face of null results based on their original hypotheses. I don't necessarily think all this is needed, but I will leave that to the editors to determine.

7. PLOS authors have the option to publish the peer review history of their article (what does this mean?). If published, this will include your full peer review and any attached files.

Reviewer #2: No

---

## [Editor Report · Acceptance letter]

14 Aug 2023

PONE-D-22-32907R2 

Get the happiness out – An online experiment on the causal effects of positive emotions on giving 

Dear Dr. Köbrich Leon:

I'm pleased to inform you that your manuscript has been deemed suitable for publication in PLOS ONE. Congratulations! Your manuscript is now with our production department. 

Kind regards, 

on behalf of

Dr. Jaume Garcia-Segarra 

Academic Editor

PLOS ONE